# Polygenic risk score for ulcerative colitis predicts immune checkpoint inhibitor-mediated colitis

Pooja Middha [1], Rohit Thummalapalli[2], Michael J. Betti [3], Lydia Yao[4], Zoe Quandt[5,6], Karmugi Balaratnam[7], Cosmin A. Bejan [8], Eduardo Cardenas[1], Christina J. Falcon[9], David M. Faleck[10], Princess Margaret Lung Group*, Matthew A. Gubens[11,12], Scott Huntsman[1], Douglas B. Johnson [13], Linda Kachuri [14,15], Khaleeq Khan[7], Min Li[1], Christine M. Lovly [16], Megan H. Murray [4], Devalben Patel[7], Kristin Werking[17], Yaomin Xu[4], Luna Jia Zhan[7], Justin M. Balko [13], Geoffrey Liu[7,18,19], Melinda C. Aldrich [3], Adam J. Schoenfeld [20] & Elad Ziv [1,12,21,22] ✉

Immune checkpoint inhibitor-mediated colitis (IMC) is a common adverse event of treatment with immune checkpoint inhibitors (ICI). We hypothesize that genetic susceptibility to Crohn's disease (CD) and ulcerative colitis (UC) predisposes to IMC. In this study, we first develop a polygenic risk scores for CD ($PRS_{CD}$) and UC ($PRS_{UC}$) in cancer-free individuals and then test these PRSs on IMC in a cohort of 1316 patients with ICI-treated non-small cell lung cancer and perform a replication in 873 ICI-treated pan-cancer patients. In a meta-analysis, the $PRS_{UC}$ predicts all-grade IMC ($OR_{meta}$=1.35 per standard deviation [SD], 95% CI = 1.12–1.64, $P = 2\times10^{-03}$) and severe IMC ($OR_{meta}$=1.49 per SD, 95% CI = 1.18–1.88, $P = 9\times10^{-04}$). $PRS_{CD}$ is not associated with IMC. Furthermore, $PRS_{UC}$ predicts severe IMC among patients treated with combination ICIs ($OR_{meta}$=2.20 per SD, 95% CI = 1.07–4.53, $P = 0.03$). Overall, $PRS_{UC}$ can identify patients receiving ICI at risk of developing IMC and may be useful to monitor patients and improve patient outcomes.

Immunotherapy with immune checkpoint inhibitors (ICI) has substantially improved clinical outcomes in patients with advanced cancers such as melanoma, non-small cell lung cancer (NSCLC), bladder, renal, breast, and other cancers[1–7]. ICIs block the ability of malignant cells to escape detection through immune checkpoints such as programmed cell death protein 1/programmed cell death ligand 1 (PD-1/PD-L1) or cytotoxic T-lymphocyte associated protein 4 (CTLA-4). Blockade of these checkpoints restores host immunosurveillance in some tumors by stimulating cytotoxic T-cells to induce cancer cell apoptosis[2,8–11].

Despite ICIs being a paradigm-shifting breakthrough in cancer treatment, enhanced activation of the immune system can lead to immune-related adverse events (irAEs) that can result in permanent discontinuation of ICIs, severe morbidity, and even patient death[12–14].

The most severe irAEs include hypophysitis, diabetes, colitis, hepatitis, and pneumonitis, with other common irAEs including rash and thyroiditis[12,15–17]. The incidence of immune checkpoint inhibitor-mediated colitis (IMC) ranges from 1%-25% and varies by ICI therapy[18,19]. The incidence of IMC is higher in patients treated with combined anti-PD-1/PD-L1 and anti-CTLA4 therapy[20,21]. Nearly 15-20% of patients receiving combination therapy develop severe IMC, which is the leading cause of hospitalization and treatment cessation[13,14,18,20,21]. Endoscopic and histological findings suggest that the presentation of IMC mimics autoimmune colitis such as ulcerative colitis (UC), a form of inflammatory bowel disease (IBD)[22,23]. Despite the phenotypic similarities between IBD and IMC, it is unclear if the underlying mechanism is shared or distinct.

A full list of affiliations appears at the end of the paper. *A list of authors and their affiliations appears at the end of the paper. ✉e-mail: Elad.Ziv@ucsf.edu

In this study, we sought to characterize the relationship between genetic predisposition to common types of autoimmune colitis (ulcerative colitis (UC) and Crohn's disease (CD)), and IMC in a cohort of NSCLC patients receiving ICI treatment. We first develop polygenic risk scores (PRS) for UC and CD using individuals not diagnosed with cancer at baseline in UK Biobank (UKB) and validate these PRSs in an independent dataset of cancer-free participants in Vanderbilt University Medical Center biobank (BioVU)[24]. Next, we evaluate the association between each of these PRS and the development of IMC in a cohort of patients with NSCLC receiving ICI therapy and conduct an independent replication in a cohort of patients with diverse cancer types treated with ICI therapy in BioVU[24]. We also investigate the association between human leukocyte antigen (HLA) alleles known to affect UC risk with IMC. Additionally, we examine the role of IMC and PRS for UC, and CD, respectively, on progression-free survival (PFS) and overall survival (OS).

## Results

### Patient characteristics

We analyzed data from 1316 study participants included in the GeRI cohort, which included four sites (Supplementary Table 1 and see "Methods"). The GeRI cohort comprised approximately 50% men and the mean age at lung cancer diagnosis was 65 years (+/−10.3). The study was composed of 69.5% of individuals who self-reported as White followed by 6.7% identifying as Asian, and 5.3% as Black. A small proportion (9%) received the combined anti-PD-1/PD-L1 and anti-CTLA-4 inhibitor therapy and the remainder received either anti-PD-1 or PD-L1 inhibitor monotherapy (91%). The cumulative incidence of IMC was ~4% (55 events); it was ~2% (32 events) for severe IMC in the GeRI cohort. The rates were similar across all study sites. The analytic strategy of our study is illustrated in Fig. 1.

### Development and validation of PRS for UC and CD

We used 70% of the cancer-free UKB dataset to tune parameters for PRS using LDpred2[25]. We then obtained effect estimates for the PRS for CD and UC in the remaining 30% (testing data). In the UKB testing data, the area under the receiver operating curve (AUROC) for the PRS$_{UC}$ was 0.66 (95% CI = 0.64−0.68), and the AUROC for PRS$_{CD}$ was 0.72 (95% CI = 0.69−0.74) (Supplementary Fig. 1). In the adjusted model, PRS$_{UC}$ was strongly associated with UC with an odds ratio (OR) of 1.84 per standard deviation (SD) (95% CI = 1.76−1.93, $p < 1.0 \times 10^{-12}$). Similarly, PRS$_{CD}$ was positively associated with CD with OR of 1.83 per SD (95% CI = 1.72−1.95, $p < 1.0 \times 10^{-12}$). We observed an intermediate correlation between the two PRSs (Pearson correlation = 0.38). Additionally, the AUROC for PRS$_{UC}$ on CD was 0.58 (95% CI: 0.57−0.60), while PRS$_{CD}$ on UC yielded an AUROC of 0.58 (95% CI: 0.56−0.59). These

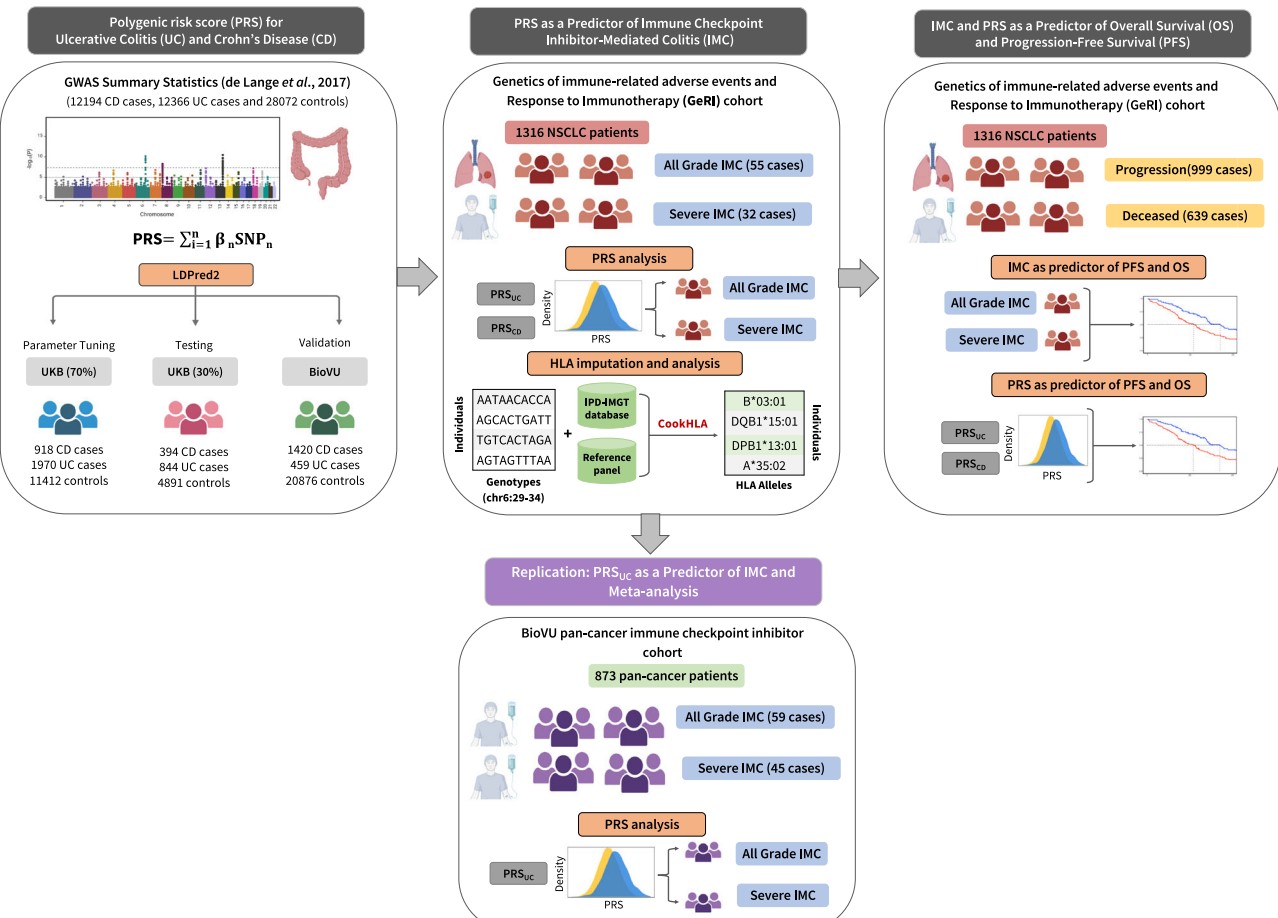

Fig. 1 | Overview of the analytical pipeline. Development and validation of the polygenic risk scores (PRSs) for ulcerative colitis and Crohn's disease was conducted in cancer-free individuals using UK Biobank and BioVU. LDpred2 method was used to tune the parameters for the PRS for ulcerative colitis and Crohn's disease (PRS$_{UC}$, PRS$_{CD}$) in 70% of the UK Biobank, using the summary statistics from the largest genome-wide association study of UC and CD. The PRSs were then tested in the remaining 30% of the UK Biobank and validated in BioVU. In the next step, the role of PRS$_{UC}$ and PRS$_{CD}$ on all-grade and severe immune checkpoint inhibitor-mediated colitis (IMC) was evaluated in a cohort of 1316 non-small cell lung cancer patients who received at least one dose of immune checkpoint inhibitor therapy. Furthermore, replication was conducted using 873 pan-cancer patients treated with immune checkpoint inhibitors obtained from BioVU. Finally, associations of all-grade and severe IMC along with PRS$_{UC}$ and PRS$_{CD}$ on progression-free survival (PFS) and overall survival (OS) were assessed. Figure created with BioRender.com.

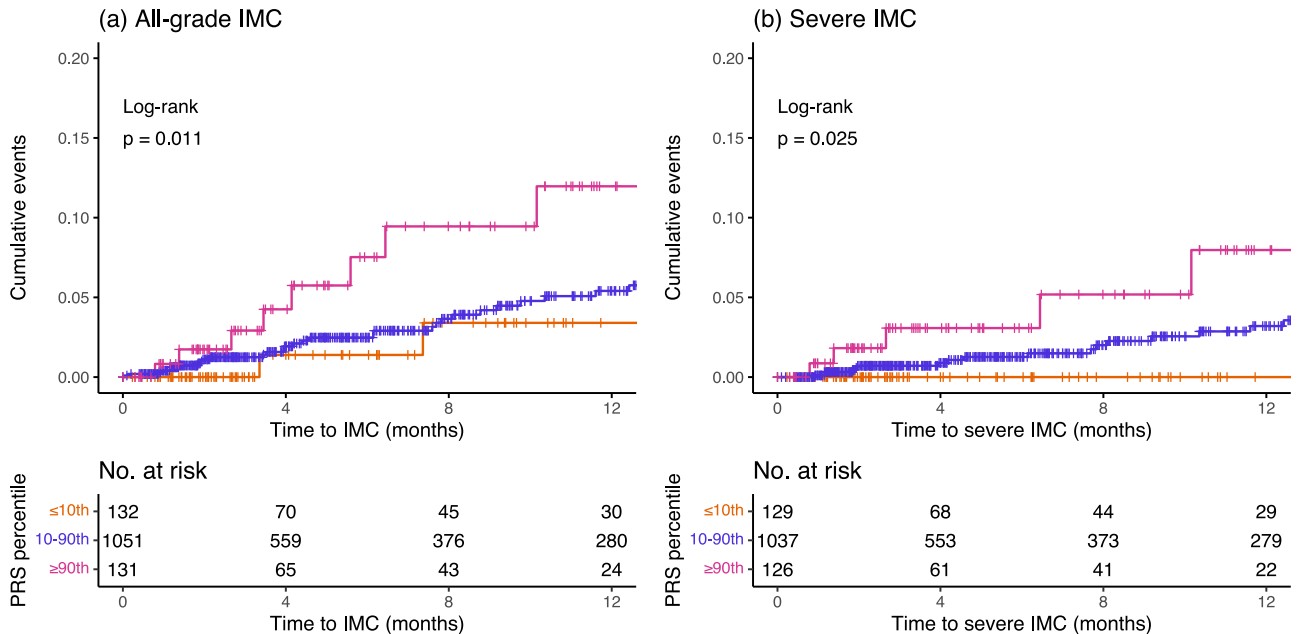

**Fig. 2 | Cumulative incidence curves of all-grade and severe immune check-point inhibitor-mediated colitis by polygenic risk score of ulcerative colitis in the GeRI cohort.** Cumulative incidence curves of **a** All-grade immune checkpoint inhibitor-mediated colitis (IMC) and **b** Severe IMC by categories of polygenic risk score of ulcerative colitis ($PRS_{UC}$) in the entire GeRI cohort. Cumulative incidence curves are unadjusted, and $PRS_{UC}$ is categorized as ≤10th percentile (low genetic risk), 10–90th percentile (average genetic risk), and >90th percentile (high genetic risk). The *p*-values included on each plot are the results of a log-rank test for the difference between the curves (two-sided). Underneath each set of curves is the number of study participants at risk beyond that time point for each of the PRS groups. Source data are provided as a Source Data file.

results suggest the presence of some shared genetic susceptibility between UC and CD. However, the distinct genetic factors influencing each phenotype remain the primary drivers of the individual PRS effects. The two PRSs were validated in another sample of cancer-free

individuals from BioVU[21]. Similar to the UKB results, the individual $PRS_{CD}$ and $PRS_{UC}$ were also strongly associated with CD and UC in BioVU. We observed an OR of 2.18 per SD (95% CI = 2.05–2.32, $p < 1.0 \times 10^{-12}$) for $PRS_{CD}$, and an OR of 1.75 per SD (95% CI = 1.59–1.92, $p < 1.0 \times 10^{-12}$) for $PRS_{UC}$. The AUROC for $PRS_{CD}$ and $PRS_{UC}$ were 0.72 (95% CI = 0.70–0.73) and 0.65 (95% CI = 0.62–0.68), respectively (Supplementary Fig. 2).

### PRS of autoimmune colitis as a predictor of IMC

The mean $PRS_{UC}$ was significantly higher in patients who developed IMC (Supplementary Fig. 3). We examined the cumulative incidence of IMC (all-grade and severe) in the top 10th percentile (high genetic risk), 10–90th percentile (average genetic risk), and lowest 10th percentile (low genetic risk) of the $PRS_{UC}$. Individuals in the top 10th percentile of the $PRS_{UC}$ had higher rates of IMC (all-grade: $p = 0.01$ and severe: $p = 0.03$) compared to the other two categories (Fig. 2). Using the Cox proportional hazards model and adjusting for genetic ancestry, recruiting site, age, sex, cancer histology, and type of therapy, we observed that the $PRS_{UC}$ was significantly associated with any diagnosis of IMC in the GeRI cohort with a hazard ratio (HR) of 1.34 per SD (95% CI = 1.02–1.76, $p = 0.04$). For a diagnosis of severe IMC, the HR per SD was 1.62 (95% CI = 1.12–2.35, $p = 0.01$) (Table 1). We found no significant association between $PRS_{CD}$ and IMC or severe IMC (Table 1).

Additionally, we conducted stratified analysis by type of therapy and histology of lung cancer to further characterize the association between $PRS_{UC}$ and IMC (all-grade and severe). For all-grade IMC, the results showed little attenuation and nominal significance when stratified by type of therapy (Table 1). However, for severe IMC we observed an HR per SD of 1.51 (95% CI = 1.01–2.27, $p = 0.04$) in patients receiving anti-PD1/anti-PD-L1 monotherapy versus a HR per SD of 4.31 (95% CI = 1.08–17.24, $p = 0.03$) in those patients receiving a combined therapy. Patients with adenocarcinoma had an HR per SD of 1.43 (95% CI = 1.06–1.93, $p = 0.02$) for all-grade IMC and an HR per SD of 2.12 (95% CI = 1.37–3.26, $p = 6 \times 10^{-04}$) for severe IMC. We also performed association analyses between ulcerative colitis PRS and IMC using different

**Table 1 | Polygenic risk score (PRS) of ulcerative colitis (UC) and Crohn's disease (CD) as a predictor of time to development of all-grade and severe immune checkpoint inhibitor-mediated colitis (IMC) in the entire GeRI cohort, using Cox proportional hazards models and stratified analysis assessing the association between $PRS_{UC}$ and all-grade/severe IMC by type of therapy and lung cancer histology**

| PRS[a] | All-grade IMC | | | Severe IMC | | |
|---|---|---|---|---|---|---|
| | HR per SD | 95% CI | *p* | HR per SD | 95% CI | *p* |
| $PRS_{UC}$ | **1.34** | **1.02–1.76** | **0.04** | **1.62** | **1.12–2.35** | **0.01** |
| $PRS_{CD}$ | 0.97 | 0.72–1.32 | 0.87 | 0.99 | 0.66–1.46 | 0.94 |
| Stratified analysis (Restricted to $PRS_{UC}$) | | | | | | |
| Therapy[b] | All-grade IMC | | | Severe IMC | | |
| Anti-PD1/Anti-PD-L1 monotherapy | 1.33 | 0.99–1.78 | 0.06 | **1.51** | **1.01–2.27** | **0.04** |
| Anti-PD1/Anti-PD-L1 + Anti-CTLA4 | 1.64 | 0.67–4.03 | 0.28 | **4.31** | **1.08–17.24** | **0.03** |
| Histology[c] | All-grade IMC | | | Severe IMC | | |
| Adenocarcinoma | **1.43** | **1.06–1.93** | **0.02** | **2.12** | **1.37–3.26** | **6×10⁻⁰⁴** |
| Squamous cell carcinoma | 0.79 | 0.16–3.78 | 0.76 | 0.79 | 0.16–3.78 | 0.76 |

All *p*-values are two-sided.
Statistically significant results are highlighted in bold.
*PRS* polygenic risk score, *IMC* immune checkpoint inhibitor-mediated colitis, *HR* hazard ratio, *SD* standard deviation, *CI* confidence interval, *UC* ulcerative colitis, *CD* Crohn's disease.
[a]Models are adjusted for age at diagnosis, sex, histology, type of therapy, recruiting site, and 5 principal components.
[b]Models are adjusted for age at diagnosis, sex, histology, recruiting site, and 5 principal components.
[c]Models are adjusted for age at diagnosis, sex, type of therapy, recruiting site, and 5 principal components.

**Table 2 | Polygenic risk score (PRS) of ulcerative colitis (UC) as a predictor of all-grade and severe immune checkpoint inhibitor-mediated colitis (IMC) in the replication cohort (BioVU) and meta-analysis (GeRI and BioVU), using logistic regression model and stratified analysis assessing the association between $PRS_{UC}$ and all-grade/severe IMC by type of therapy**

| $IMC^a$ | Replication cohort BioVU | | | Meta-analysis GeRI+ BioVU | | |
|---|---|---|---|---|---|---|
| | OR per SD | 95% CI | p | OR per SD | 95% CI | p |
| All-grade | 1.29 | 0.98–1.69 | 0.07 | **1.35** | **1.12–1.64** | **$2 \times 10^{-3}$** |
| Severe | **1.39** | **1.02–1.90** | **0.04** | **1.49** | **1.18–1.88** | **$9 \times 10^{-4}$** |
| Stratified analysis by type of therapy: All-grade IMC | | | | | | |
| $Therapy^b$ | Replication cohort BioVU | | | Meta-analysis GeRI + BioVU | | |
| Anti-PD1/Anti-PD-L1 monotherapy | 1.25 | 0.88–1.78 | 0.21 | **1.35** | **1.07–1.69** | **0.01** |
| Anti-PD1/Anti-PD-L1 + Anti-CTLA4 | 2.04 | 0.79–5.28 | 0.14 | 1.80 | 0.95-3.41 | 0.07 |
| Anti-CTLA4 monotherapy | 0.92 | 0.67–1.26 | 0.59 | - | - | - |
| Stratified analysis by type of therapy: Severe IMC | | | | | | |
| $Therapy^b$ | Replication cohort BioVU | | | Meta-analysis GeRI + BioVU | | |
| Anti-PD1/Anti-PD-L1 monotherapy | 1.47 | 0.96–2.25 | 0.08 | **1.48** | **1.10–1.98** | **$9 \times 10^{-3}$** |
| Anti-PD1/Anti-PD-L1 + Anti-CTLA4 | 1.89 | 0.74–4.86 | 0.19 | **2.20** | **1.07–4.53** | **0.03** |
| Anti-CTLA4 monotherapy | 1.00 | 0.71–1.40 | 0.99 | - | - | - |

All p-values are two-sided.
Statistically significant results are highlighted in bold.
*IMC* immune checkpoint inhibitor-mediated colitis, *OR* odds ratio, *SD* standard deviation, *CI* confidence interval.
$^a$Models are adjusted for age at diagnosis, sex, type of therapy, and 5 principal components.
$^b$Models are adjusted for age at diagnosis, sex, and 5 principal components.

**Table 3 | All-grade and severe immune checkpoint inhibitor-mediated colitis (IMC) as predictors of progression-free survival (PFS) and overall survival (OS) in the entire GeRI cohort, using Cox proportional hazards models with 90-day landmark**

| IMC | PFS | | | OS | | |
|---|---|---|---|---|---|---|
| | HR | 95% CI | p-value | HR | 95% CI | p-value |
| All-grade | 0.80 | 0.55–1.17 | 0.26 | **0.40** | **0.24–0.66** | **$3 \times 10^{-04}$** |
| Severe | 0.61 | 0.34–1.09 | 0.09 | **0.23** | **0.09–0.55** | **$9 \times 10^{-04}$** |

All p-values are two-sided.
Statistically significant results are highlighted in bold.
*IMC* immune checkpoint inhibitor-mediated colitis, *PFS* progression-free survival, *OS* overall survival, *HR* hazards ratio, *CI* confidence interval.
All models are adjusted for age at diagnosis, sex, histology, type of therapy, recruiting site, and 5 principal components.

previously published $PRS_{UC}$ and noted consistent and robust trends toward the association (Supplementary Table 2).

## Replication of the association between $PRS_{UC}$ and IMC
Replication was conducted within an independent study of 873 patients from a pan-cancer cohort in BioVU[24] who underwent treatment with either anti-PD1/PD-L1 monotherapy or combination ICI therapy. The characteristics of the replication cohort are shown in Supplementary Table 2. Briefly, the replication study consisted of 63% males and 37% females. Among 873 ICI-treated patients, approximately 95% of the patients received anti-PD1/PD-L1 monotherapy and 5% of

patients received combined anti-PD-1/PD-L1 and anti-CTLA4 therapy. An additional 274 cancer patients were identified and were treated with anti-CTLA4 monotherapy.

The results from the analysis in the replication study are presented in Table 2. In our analysis of 873 patients, we found a trend toward association between $PRS_{UC}$ and all-grade IMC (OR per SD = 1.29, 95% CI = 0.98–1.69, p = 0.07). However, for $PRS_{UC}$ and severe IMC, we observed statistically significant replication with an OR per SD of 1.39 (95% CI = 1.02–1.90, p = 0.04, Table 2). Within our stratified analysis by type of therapy, for anti-PD1/PD-L1 monotherapy, we observed an OR per SD of 1.25 (95% CI = 0.88–1.78, p = 0.21) for all-grade IMC, while a slightly stronger and nominally significant association was seen for severe IMC (OR per SD = 1.47, 95% CI = 0.96–2.25, p = 0.08). For those receiving dual therapy, we observed an OR per SD of 2.04 (95% CI = 0.79–5.28, p = 0.14) for all-grade IMC and an OR per SD of 1.89 (95% CI = 0.74–4.86, p = 0.19) for severe IMC. Furthermore, we conducted an adjusted logistic regression model within the anti-CTLA4 monotherapy (N = 274) and found an OR per SD of 0.92 (95% CI = 0.67–1.26, p = 0.59) for all-grade IMC. For severe IMC in the anti-CTLA4 monotherapy group, we observed an OR per SD of 1.00 (95% CI = 0.71–1.40, p = 0.99).

## Meta-analysis of $PRS_{UC}$ and IMC associations in discovery and replication studies
Next, we performed a meta-analysis using fixed-effect inverse-variance weighting, combining the logistic regression models from the initial GeRI cohort and BioVU replication cohort (Table 2). Our findings show a significantly positive association between $PRS_{UC}$ and all-grade IMC with an $OR_{meta}$ per SD of 1.35 (95% CI = 1.12–1.64, p = $2 \times 10^{-03}$). Similarly, a robust association of $PRS_{UC}$ and severe IMC was observed with an $OR_{meta}$ per SD of 1.49 (95% CI = 1.18–1.88, p = $9 \times 10^{-04}$).

For patients who received anti-PD1/PD-L1 monotherapy, $PRS_{UC}$ demonstrated a significant association with all-grade IMC, showing an $OR_{meta}$ per SD of 1.35 (95% CI = 1.07–1.69, p = 0.01). Similarly, a stronger association was observed with severe IMC, with an $OR_{meta}$ per SD of 1.48 (95% CI = 1.10–1.98, p = $9 \times 10^{-3}$). Among patients treated with combination or dual therapy, a trend towards association with all-grade IMC was seen ($OR_{meta}$ per SD = 1.80, 95% CI = 0.95–3.41, p = 0.07); however, a robust and pronounced association was found in relation to severe IMC ($OR_{meta}$ per SD = 2.20, 95% CI = 1.07–4.53, p = 0.03).

## Role of known UC-HLA associations on IMC in GeRI cohort
We assessed the association between all-grade IMC and HLA markers known to be associated with ulcerative colitis[26,27] (Supplementary Fig. 4). Out of 12 known UC-associated HLA markers, we observed an OR of 2.63 (95% CI = 1.08–6.40, p = 0.03) for HLA-DRB1*12:01 and all-grade IMC. However, at a false-discovery rate (FDR) < 0.05 none of the known HLA markers were associated with all-grade IMC in the GeRI cohort.

## IMC and PRS of autoimmune colitis as a predictor of PFS and OS
To assess the role of IMC on clinical outcomes, we conducted a Cox proportional hazards model with a 90-day treatment landmark in the GeRI cohort (Table 3 and Fig. 3). We observed the effect of all-grade IMC on OS with an HR of 0.40 (95% CI = 0.24–0.66, p = $3.0 \times 10^{-04}$) and of severe IMC on OS with an HR of 0.23 (95% CI = 0.09–0.55, p = $9.0 \times 10^{-04}$). However, we observed no significant association between PFS and IMC (Table 3 and Supplementary Fig. 5).

Despite the association between $PRS_{UC}$ and IMC, $PRS_{UC}$ was not associated with PFS (HR per SD = 1.00, 95% CI = 0.94–1.07, p = 0.99) and OS (HR per SD = 1.01, 95% CI = 0.93–1.09, p = 0.91) in the GeRI cohort (Table 4). Similarly, we observed no association between $PRS_{CD}$ and PFS (HR per SD = 0.98, 95% CI = 0.91–1.05, p = 0.50) and OS (HR per SD = 1.02, 95% CI = 0.93–1.11, p = 0.68), respectively (Table 4).

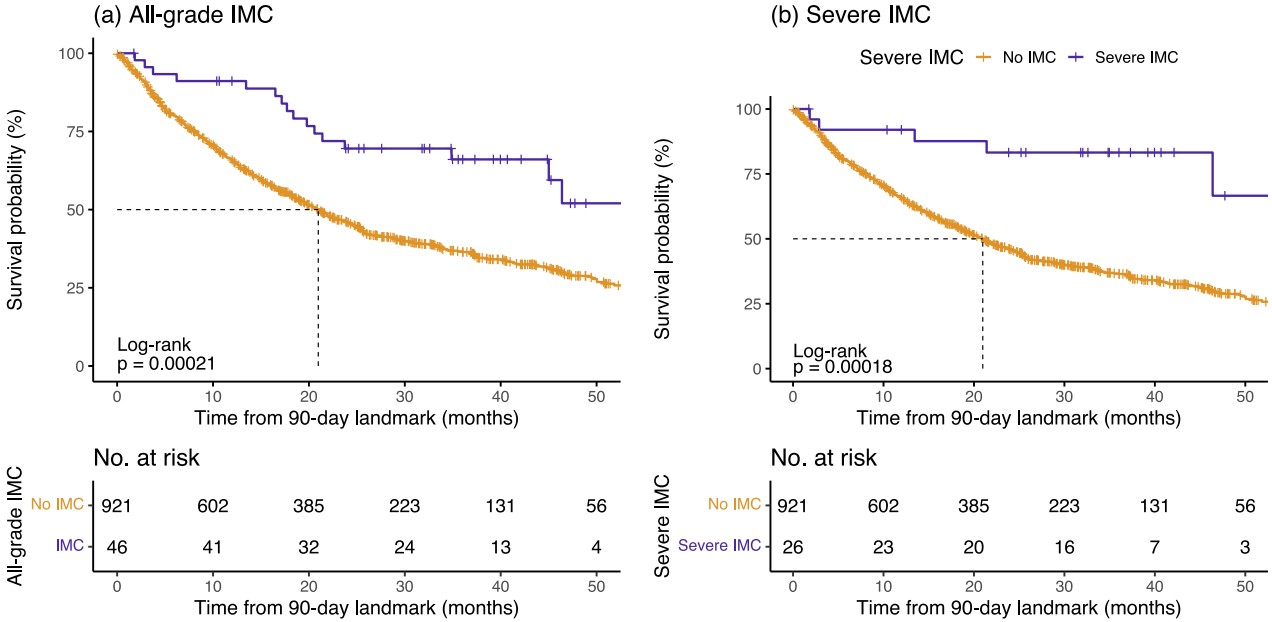

**Fig. 3 | Immune checkpoint inhibitor-mediated colitis (IMC) as a predictor of overall survival (OS) in the entire GeRI cohort. a** All-grade IMC and **b** Severe IMC. Kaplan–Meier survival curves are unadjusted with 90-day landmark and compare those who had an IMC (all-grade or severe) with those who did not have an IMC (No IMC). The *p*-values in the graph represent the log-rank p-values (two-sided), and the dotted line represents the median survival time. Underneath each set of curves is the number of study participants at risk beyond that time point for the IMC and No IMC groups. Source data are provided as a Source Data file.

## Discussion

Immune checkpoint inhibitors are part of standard regimens to treat many advanced cancers and are used in adjuvant and neoadjuvant settings for early-stage diseases in multiple cancers[3,4,10,28–32]. Immune-related adverse events are common complications from ICI, and there are few predictors of irAEs[33,34]. We sought to identify genetic predictors of immune checkpoint inhibitor-mediated colitis which frequently results in hospitalization and ICI discontinuation and can occasionally lead to death[18,19,35]. Specifically, we evaluated the relationship between genetic predisposition for autoimmune colitis (UC, CD) and IMC, and found that the PRS_UC can predict IMC. The association was stronger when analyses were restricted to individuals with severe IMC—an important finding as the most important clinical cases to identify were best predicted by PRS_UC. Furthermore, we investigated the role of HLA markers associated with UC on the development of IMC. However, we did not have HLA typing for these individuals, and therefore, the imputation of HLA was not validated. Furthermore, our study was not well-powered to detect the effect of many different HLA alleles after multiple hypothesis testing. Future studies will need to analyze HLA effects on IMC.

Our findings significantly contribute to our understanding of the biological underpinnings of IMC and may also impact the management of patients treated with ICIs. First, we demonstrate that IMC has some genetic overlap with UC, but we found no evidence for overlap with CD. This is notable despite the correlation observed in our PRS for UC and CD, signifying that the genetic factors associated with IMC align more closely with the distinct genetic markers associated with UC. Our finding is also consistent with clinical reports in which the most frequent phenotype of IMC resembles UC most closely[22,23,36]. Our results also suggest that as the genetic risk prediction of UC improves, the genetic risk of IMC may also be improved. In particular, rare variants in certain genes substantially increase the risk of UC and we hypothesize may also affect IMC risk[37–39]. Prior reports on ICI-induced hypothyroidism[40,41] and rash[42] demonstrated that PRS for autoimmune disorders predict irAEs, suggesting that ICI may unmask autoimmune syndromes in some genetically predisposed individuals.

We also found that individuals who developed IMC had improved survival outcomes when compared to those who did not develop IMC, including in a landmark sensitivity analysis, which is concordant with previously published literature[43–48]. However, PRS_UC and PRS_CD were not associated with PFS or OS, suggesting that the genetic basis of autoimmune disease susceptibility is distinct from genetic factors influencing survival outcomes. It has been postulated that both anti-tumor responses to ICIs, and the development of irAEs are representative of a robust immune response; however, one possible explanation for our finding is that the genetic contributions captured in the autoimmune PRSs are probably capturing the cross-presentation of shared antigens which may not be associated with clinical outcomes. This suggests there could be other genetic and environmental factors driving the association between IMC and overall survival.

Our study has several implications that may impact the care of cancer patients treated with ICIs. For example, our results suggest that germline genotyping could help assist selection of patients at high risk of IMC in a clinical trial setting to assess the role of preventative measures such as the commencement of concurrent anti-TNFα therapies or anti-integrin α4β7 antibodies[49,50] along with ICI treatment in patients at high risk for IMC and toxicity-related early

**Table 4 | Polygenic risk scores of ulcerative colitis (PRS_UC) and Crohn's disease (PRS_CD) as predictors of progression-free survival (PFS) and overall survival (OS) in the GeRI cohort, using Cox proportional hazards models**

| PRS | PFS | | | OS | | |
|---|---|---|---|---|---|---|
| | HR per SD | 95% CI | *p*-value | HR per SD | 95% CI | *p*-value |
| PRS_UC | 1.00 | 0.94–1.07 | 0.99 | 1.01 | 0.93–1.09 | 0.91 |
| PRS_CD | 0.98 | 0.91–1.05 | 0.50 | 1.02 | 0.93–1.11 | 0.68 |

All *p*-values are two-sided.
All models are adjusted for age at diagnosis, sex, histology, type of therapy, recruiting site, and 5 principal components.
*PRS* polygenic risk score, *PFS* progression-free survival, *OS* overall survival, *HR* hazards ratio, *SD* standard deviation, *CI* confidence interval, *UC* ulcerative colitis, *CD* Crohn's disease.

treatment cessation. Additionally, these findings may also help facilitate clinical decision-making. Combination immunotherapies are more effective but are also associated with a substantially increased risk of irAEs[45,51–55]. Our stratified analysis by type of therapy demonstrated the association between PRS$_{UC}$ and severe IMC in individuals receiving anti-PD-1/PD-L1 and anti-CTLA4 combination therapy. Among patients who may be candidates for combination immunotherapies but have a high genetic risk based on PRS$_{UC}$, oncologists may consider monotherapy, particularly in clinical situations in which the benefits of dual therapy on disease control may be modest. Conversely, patients who are at relatively low risk based on PRS$_{UC}$ may be better candidates for combination therapy. In addition, the use of PRS$_{UC}$ might also be considered to assist with treatment decisions in clinical settings where ICI therapy is approved but there is substantial clinical equipoise; for example, in the adjuvant setting for patients with resectable NSCLC[56,57] and low PD-L1 expression or adjuvant setting for resected stage II melanoma[58]. Our analysis within the anti-CTLA4 monotherapy subgroup did not reveal any significant association between PRS$_{UC}$ and IMC. These results should be interpreted cautiously since the sample size was limited in this subgroup. However, anti-CTLA4 as monotherapy has become less common in contemporary clinical practice, with its predominant use being in combination with anti-PD1/PD-L1 therapy, and our PRS$_{UC}$ did predict IMC in these patients. Our initial findings were observed in a cohort of NSCLC patients. However, our replication study included a broader array of pan-cancer studies and demonstrated the generalizability of PRS$_{UC}$ to predict IMC.

Although our study has important clinical implications and strengths, it also has some limitations. While PRS effectively captures established variants associated with UC, it may not account for unidentified genetic contributors (missing heritability). Nevertheless, as we unveil the missing heritability of UC, we expect to further improve the polygenic prediction of IMC. Furthermore, we developed these PRSs in a predominantly European ancestry cohort (UK Biobank) and the GeRI cohort and BioVU replication study were also predominantly of European ancestry; more work is needed to generalize these results to other ancestries. In addition, there may be other limitations to implementing PRS in the clinic including cost, rapidity of return of results, and reliability and consistency across different algorithms[59–62]. Although we included one replication cohort, additional studies of more ICI-treated patients will help strengthen our findings and, in particular, may give better power to evaluate HLA associations and other individual loci that may improve our understanding of the genetic similarities and potential differences between IMC and UC. For most complex traits, including autoimmune disorders and, likely for irAEs, environmental factors also play an important role. Our study does not address how environmental factors affect the risk of IMC. For example, the gut microbiome may modify susceptibility to and severity of IBD[63] and, therefore, may also contribute to the susceptibility of developing IMC in cancer patients who have undergone ICI treatment. To determine the joint associations between PRS$_{UC}$ and environmental risk factors, further studies are necessary.

We also found an association between IMC and OS. This result could be due to survivor bias[64,65], where patients who respond to therapy and are on therapy longer are at an increased risk of developing irAEs. We used a 90-day landmark analysis[66] to account for this bias for both PFS and OS, although this may not completely eliminate the survivor bias.

Overall, our findings suggest a shared genetic basis between ulcerative colitis and immune checkpoint inhibitor-mediated colitis among patients undergoing ICI treatment. Prediction of IMC using genetic information should create new opportunities for better risk stratification and ultimately for better management and possibly prevention of this common and important side effect from immunotherapy.

## Methods

This research complies with all relevant ethical regulations. Institutional Review Board approvals were obtained at each site individually, and written informed consent was acquired from all study participants prior to inclusion in the study.

### Study population

Genetics of immune-related adverse events and Response to Immunotherapy (GeRI) cohort is comprised of 1316 advanced Stage IIIB/IV NSCLC patients who received ICI therapy (PD-1 or PD-L1 inhibitors as monotherapy or in combination with either CTLA-4 inhibitors and/or chemotherapy) and were recruited from four different institutions: Memorial Sloan Kettering Cancer Center (MSKCC), Vanderbilt University Medical Center (VUMC), Princess Margaret Cancer Center (PM), and University of California, San Francisco (UCSF).

A total of 752 individuals were treated with ICIs at MSKCC between 2011 and 2018 and had an available blood sample. Clinical data were extracted from a manual review of medical and pharmacy records for demographics, lung cancer histology, and ICI treatment history, including detailed information on immune-related adverse events (irAEs). The VUMC cohort is composed of 267 patients who received ICI therapy at the medical center between 2009 and 2019. Patients participated in BioVU[21], Vanderbilt's biomedical repository of DNA that is linked to de-identified health records. Treatment dates and irAEs were extracted using manual chart review by a trained thoracic oncology nurse. The PM cohort included 266 advanced NSCLC patients who received ICI therapy between 2011 and 2022; all provided a blood sample and completed a questionnaire. Clinical data were manually extracted by trained abstractors and supplemented by the PM Cancer Registry. From UCSF, 31 patients who had received ICIs were identified by thoracic oncologists between 2019 and 2021 and provided either a blood or saliva sample after informed consent. Clinical data including, demographics, history of lung cancer and ICI treatment, and irAEs were extracted after a manual review of electronic health records.

### Immune checkpoint inhibitor-mediated colitis (IMC)

After the initiation of ICI therapy, immune checkpoint inhibitor-mediated colitis (IMC) was defined based on clinical chart review and documentation of IMC by the primary oncologist, gastroenterologist, and/or other clinicians treating the patient based on clinical features and/or radiologic/histologic evidence suggesting colitis due to ICI. Participants who were diagnosed with infectious causes of colitis including *Clostridium difficile*, or a pathogen on a gastrointestinal pathogen panel or ova and parasite test were excluded. To assess the severity of IMC, we used 2 metrics based on NCI Common Terminology Criteria for Adverse Events Version 5 (NCI-CTCAE) that capture grade 3 IMC or above (i) hospitalization for management of IMC and/or (ii) permanent cessation of ICI therapy due to the adverse event.

IMC was coded as a dichotomous variable (1: all IMC, 0: no IMC) and time-to-IMC was assessed from the start of the ICI therapy to the date of onset of IMC or the date of ICI discontinuation due to IMC. Patients who did not experience IMC were censored either at the end of treatment due to any reason or last follow-up date if the treatment was ongoing. Based on the severity criteria, severe IMCs were also coded as binary variables (1: severe IMC, 0: no IMC).

### Ascertainment of clinical outcomes

Progression-free survival (PFS) and overall survival (OS) were evaluated from the date of initiation of ICI therapy to the date of progression and death, respectively, at MSK, PM, and UCSF sites. At VUMC, time-to-discontinuation of therapy due to progression from therapy initiation was used as a surrogate. If the treatment was ongoing, patients were censored at the date of the last follow-up. The VUMC

cohort is de-identified and not linked to the National Death Index; therefore, all-cause mortality (overall survival) information is unavailable for VUMC participants ($n = 267$).

### Quality control, genotyping, and imputation of the GeRI cohort

DNA from blood or saliva was extracted and genotyped using Affymetrix Axiom Precision Medicine Diversity Array. Samples with a call rate <95% were excluded from the analysis and SNPs with missing rates >5% were also excluded from the analysis. Genetic ancestry was calculated using principal component analysis in PLINK after linkage disequilibrium pruning ($R^2 < 0.1$). Imputation was performed using the Michigan Imputation Server with the 1000 Genomes phase3 v5 reference panel. Standard genotyping and quality control procedures were implemented. Variants with minor allele frequency <0.01 were excluded from the analysis.

### Development and validation of polygenic risk score (PRS) for autoimmune colitis

We developed PRS for CD (1312 CD cases and 16,303 controls) and UC (2814 UC cases and 16,303 controls), separately using UK Biobank (UKB) data, where we divided the data into two parts: 70% for hyperparameter tuning and 30% of the remaining data for testing the PRS. Genetic data from both the UKB Affymetrix Axiom array (89%) or the UK BiLEVE array (11%)[67] which have been imputed using the Haplotype Reference Consortium and the UK10K and 1000 Genomes phase3 reference panels[67] were utilized in the analysis. Analyses were restricted to European ancestry individuals based on self-reported White ethnicity and genetic ancestry PCs within five standard deviations of the population mean. Samples with discordant self-reported and genetic sex were excluded. Additionally, we also excluded one sample from each pair of first-degree relatives. Samples with greater than five standard deviations from the mean heterozygosity were further excluded from the analysis. Information from both self-report and ICD9/10 codes were used to capture CD (1312 cases) and UC (2814 cases) phenotypes in UKB.

We used the LDPred2[25] method to develop PRS of CD and UC. LDpred2 estimates the posterior effect sizes based on summary statistics from genome-wide association studies while taking into account the linkage disequilibrium between variants and assuming a prior on the markers. To derive PRS, summary statistics were obtained from the previously published largest genome-wide association study of CD, and UC[68]. We restricted the analysis to HapMap3 variants and implemented LDPred2-auto function to evaluate the posterior effect sizes for each variant. LDPred2-auto first estimates the proportion of causal variants and heritability for the trait under evaluation. Next, it determines the posterior effects estimates for the included variants. The final PRS weights are available at PGS catalog (See Data Availability). Briefly, $PRS_{UC}$ included 744,575 variants, whereas $PRS_{CD}$ comprised 744,682 variants.

PRS was constructed using the formula: $PRS = \beta_1 \times SNP_1 + \beta_2 \times SNP_2 + \ldots\ldots + \beta_n \times SNP_n$, where $\beta$ was estimated using LDPred2-auto function. Each PRS was standardized to have a mean of zero and a standard deviation of 1. The association of $PRS_{CD}$ and $PRS_{UC}$ with each respective target phenotype was assessed using logistic regression models, adjusted for age at diagnosis for cases and age at enrollment for controls, sex, genotyping array, and the top 10 genetic ancestry principal components (PCs). Area under the receiver operating characteristic (AUROC) curves were calculated in the testing dataset and used to assess the overall prediction accuracy of each PRS in UKB.

We validated the two PRSs in a sample of cancer-free individuals (1420 CD cases, 459 UC cases, and 20,876 controls in the VUMC BioVU[24]. All analyses were restricted to individuals of European ancestry and adjusted for age, sex, and ten principal components. AUROC curves were used to estimate the prediction of the PRSs.

### Assessment of autoimmune colitis PRS to predict IMC in GeRI cohort

Using the weights generated from LDPred2 for CD, and UC, we separately calculated two weighted PRSs ($PRS_{CD}$, $PRS_{UC}$) for the GeRI participants. The cumulative incidence of IMC (all-grade and severe) was assessed by categories of PRS percentiles. Individuals in the top 10% of the PRS distribution (PRS > 90th percentile) were classified as having high genetic risk, those in the bottom 10% (PRS ≤ 10th percentile) were classified as low risk, and the middle category (>10th to ≤90th percentile) classified as average genetic risk. Additionally, to evaluate the performance of each potential PRS on either time-to-IMC or time-to-severe IMC, we used Cox proportional hazards models, adjusted for age at diagnosis, sex, lung cancer histology, type of therapy, recruiting site, and the first 5 genetic ancestry PCs. To further understand the differential effects of type of therapy and histology on the association between $PRS_{UC}$ and IMC, we conducted stratified analysis by type of ICI therapy and histology of lung cancer.

### Replication of $PRS_{UC}$ and IMC in an independent study

We performed an independent replication to further characterize the association between $PRS_{UC}$ and IMC. Our replication study comprises of 873 patients enrolled in BioVU[24], across all cancer types and treated with either anti-PD-1/PD-L1 monotherapy or a combination of anti-PD1/PD-L1 and anti-CTLA4 therapy. There was no overlap of samples between individuals from BioVU included in the GeRI cohort (discovery) and the replication dataset from BioVU. Immune checkpoint inhibitor-mediated colitis was ascertained by a manual review of the electronic health records. An IMC case was defined as either biopsy-confirmed colitis or the occurrence of diarrhea in ICI patients, not attributable to any other cause, that required treatment with steroids and subsequently showed improvement with steroid therapy. All samples were genotyped using Illumina Expanded Multi-Ethnic Genotyping Array (MEGA-EX) and imputed to 1000 Genomes reference panel (version 3)[24]. Post-imputation standard quality control procedures were employed to exclude low-quality variants and samples. In short, samples with a call rate <95% and SNPs with missing rates >2% were excluded from the analysis. Additionally, all SNPs with minor allele frequency <1% and Hardy–Weinberg $p$-value < 1e-06, and INFO < 0.95 were excluded.

We performed unconditional logistic regression to assess the association between $PRS_{UC}$ and all-grade IMC and severe IMC, respectively. All models were adjusted for age at diagnosis, sex, type of therapy, and 5 principal components. In addition, we conducted stratified logistic regression by type of therapy, and the models were adjusted for age at diagnosis, sex, and 5 principal components. This study had an additional 274 patients who received anti-CTLA4 monotherapy, and we further evaluated the association between $PRS_{UC}$ and IMC separately in this group.

### Meta-analysis of the association between $PRS_{UC}$ and IMC

For meta-analysis, we conducted standard logistic regression adjusted for age at diagnosis, sex, type of therapy, site, and 5 PCs in the GeRI study. Next, we carried out an inverse-variance weighted fixed-effect meta-analysis between our discovery and replication studies. Additionally, we conducted a meta-analysis of the stratified results by type of therapy in the GeRI cohort and the replication study from BioVU.

### Role of HLA markers associated with UC and CD on IMC in GeRI cohort

To elucidate the role of known UC-associated HLA markers on IMC, we performed HLA imputation using CookHLA[69] and HATK[70]. HLA alleles were imputed at 2-field resolution against the Type 1 Diabetes Genetics Consortium reference panel[71] and using the nomenclature from IPD-IMGT/HLA database v3.51. Association analysis with all-grade IMC was conducted using logistic regression models adjusted for age at diagnosis, sex, lung cancer histology, type of therapy, recruiting site, and 5

PCs. Analyses were restricted to common HLA alleles (frequency ≥0.01) known to be associated with ulcerative colitis[26].

## Impact of IMC and PRS of autoimmune colitis on PFS and OS in GeRI cohort

The association of IMC (all-grade and severe) on PFS and OS was examined using the Cox proportional hazards model by examining only the patients who had PFS and OS longer than 90 days (90-day landmark)[66]. All models were adjusted for age at diagnosis, sex, lung cancer histology, type of therapy, and 5 PCs. Survival curves and rates were estimated using the Kaplan–Meier method. To investigate the association between $PRS_{CD}$, $PRS_{UC}$ on PFS, and OS, we conducted Cox proportional hazards models, adjusted for age at diagnosis, sex, histology, type of therapy, and 5 PCs. All $p$-values are two-sided, and analyses were conducted using Plink2, R v4.2.2 (R Foundation for Statistical Computing) with RStudio v2022.12.0.353.

## Reporting summary

Further information on research design is available in the Nature Portfolio Reporting Summary linked to this article.

## Data availability

UK Biobank data are publicly available by request from https://www.ukbiobank.ac.uk. Scoring files for Crohn's disease and ulcerative colitis are available from the PGS catalog http://www.pgscatalog.org/score/PGS004253/ and http://www.pgscatalog.org/score/PGS004254/. De-identified data along with outcomes used for this work are available at https://github.com/PoojaMiddha/GeRI_colitis/tree/GeRI_colitis_manuscript[72]. BioVU data is available to Vanderbilt-affiliated members subject to approval by the BioVU Review Committee from https://victr.vumc.org/how-to-use-biovu/[24]. External Users will need a Vanderbilt PI for all BioVU projects, and contracts will need to be in place before data can be shared. Investigators can reach out to BioVU (biovu@vumc.org) if they would like more information or help establishing a collaboration. The remaining data are available within the Article, Supplementary Information, or Source Data file. Source data are provided in this paper.

## Code availability

We used R programming language (v4.2.2), survival R package (v3.4.0), ggplot2 package (v3.4.2), pROC package (v1.18.0), bigsnpr package (v1.11.6), and PLINK2 for PRS calculations. Analysis scripts can be found at https://github.com/PoojaMiddha/GeRI_colitis/tree/GeRI_colitis_manuscript[72].

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

## Acknowledgements

This work was supported by the National Institutes of Health R01-CA227466 and K24-CA169004 to E.Ziv; C.M.Lovly was supported in part by NIH NCI UG1CA233259, P01CA129243, and P30CA068485; R.Thummalapalli was supported by T32-CA009207; The Lusi Wong Fund, Posluns Fund, Alan Brown Chair in Molecular Genomics, Princess Margaret Cancer Foundation were awarded to G. Liu for this work; M.C. Aldrich was supported in part by R01-CA227466, U01CA253560, R01CA251758 and the Vanderbilt Institute for Clinical and Translational Research (UL1TR002243); Z. Quandt was supported NIDDK DiabDocs K12DK133995 and a Larry L Hillblom Foundation Start-Up Grant; A.J.Schoenfeld, D.Faleck were supported by the Memorial Sloan Kettering Cancer Center Support Grant/Core (P30-CA008748), the Druckenmiller Center for Lung Cancer Research at Memorial Sloan Kettering Cancer Center. Justin M. Balko was supported by NIH/NCI R01CA227481; Douglas B Johnson was supported by NIH/NCI R01CA227481 and NIH/NHLBI R01HL156021. The samples and/or dataset(s) used for the analyses described were obtained from Vanderbilt University Medical Center's BioVU which is supported by numerous sources: institutional funding, private agencies, and federal grants. These include the NIH-funded Shared Instrumentation Grant S10OD017985 and S10RR025141; and CTSA grants UL1TR002243, UL1TR000445, and UL1RR024975. Genomic data are also supported by investigator-led projects that include U01HG004798, R01NS032830, RC2GM092618, P50GM115305, U01HG006378, U19HL065962, R01HD074711; and additional funding sources listed at https://victr.vumc.org/biovu-funding/. Megan H. Murray's work on this project was completed in August 2022 while she was working at Vanderbilt University Medical Center.

## Author contributions

Concept and design: P.M., M.C.A., A.J.S., E.Z.; Data acquisition: P.M., R.T., Z.Q., K.B., C.A.B., E.C., C.J.F., P.M.L.G., M.A.G., S.H., D.J.B., L.K., K.K., M.L., C.M.L., M.H.M., D.P., K.W., Y.X., L.J.Z., J.M.B., G.L., M.C.A., A.J.S., E.Z.; Analysis or interpretation of data: P.M., R.T., M.J.B., L.Y., Z.Q., D.M.F., L.K., C.M.L., G.L., M.C.A, A.J.S., E.Z.; Drafting of the manuscript: P.M., R.T., M.J.B., L.Y., Z.Q., K.B., C.A.B., E.C., C.J.F., D.M.F., P.M.L.G, M.A.G., S.H., D.B.J., L.K., K.K., M.L., C.M.L., M.H.M., D.P., K.W., Y.X., L.J.Z., J.M.B., G.L., M.C.A., A.J.S., E.Z. Critical revision of the manuscript for important intellectual content: P.M., R.T., G.L., M.C.A., A.J.S., E.Z.

## Competing interests

J.M.B. receives research support from Genentech/Roche and Incyte Corporation, has received advisory board payments from AstraZeneca and Mallinckrodt, and is an inventor on patents regarding immunotherapy targets and biomarkers in cancer. The remaining authors declare no other competing interests.

## Additional information

¹Department of Medicine, University of California San Francisco, San Francisco, CA, USA. ²Department of Medicine, Memorial Sloan Kettering Cancer Center, New York, NY, USA. ³Department of Medicine, Division of Genetic Medicine, Vanderbilt University Medical Center, Nashville, TN, USA. ⁴Department of Biostatistics, Vanderbilt University Medical Center, Nashville, TN, USA. ⁵Division of Endocrinology and Metabolism, Department of Medicine, University of California San Francisco, San Francisco, CA, USA. ⁶Diabetes Center, University of California San Francisco, San Francisco, CA, USA. ⁷Princess Margaret Cancer Centre, Toronto, ON, Canada. ⁸Department of Biomedical Informatics, Vanderbilt University Medical Center, Nashville, TN, USA. ⁹Fiona and Stanley Druckenmiller Center for Lung Cancer Research, Memorial Sloan Kettering Cancer Center, New York, NY, USA. ¹⁰Gastroenterology, Hepatology & Nutrition Service, Department of Medicine, Memorial Sloan Kettering Cancer Center, New York, NY, USA. ¹¹Division of Hematology/Oncology, Department of Medicine, University of California San Francisco, San Francisco, CA, USA. ¹²Helen Diller Family Comprehensive Cancer Center, University of California San Francisco, San Francisco, CA, USA. ¹³Department of Medicine, Vanderbilt University Medical Center, Nashville, TN, USA. ¹⁴Department of Epidemiology and Population Health, Stanford University School of Medicine, Stanford, CA, USA. ¹⁵Stanford Cancer Institute, Stanford University of Medicine, Stanford, CA, USA. ¹⁶Department of Medicine, Division of Hematology and Oncology, Vanderbilt University Medical Center and Vanderbilt Ingram Cancer Center, Nashville, TN, USA. ¹⁷Department of Thoracic Surgery, Vanderbilt University Medical Center, Nashville, TN, USA. ¹⁸Temerty School of Medicine, Toronto, ON, Canada. ¹⁹Dalla Lana School of Public Health, University of Toronto, Toronto, ON, Canada. ²⁰Thoracic Oncology Service, Memorial Sloan Kettering Cancer Center, New York, NY, USA. ²¹Center for Genes, Environment and Health, University of California San Francisco, San Francisco, CA, USA. ²²Institute for Human Genetics, University of California San Francisco, San Francisco, CA, USA. *A list of authors and their affiliations appears at the end of the paper. ✉e-mail: Elad.Ziv@ucsf.edu

## Princess Margaret Lung Group

**Natasha B. Leighl⁷, Penelope A. Bradbury⁷, Frances A. Shepherd⁷, Adrian G. Sacher⁷ & Lawson Eng⁷**

