## [Peer Review File · Nature Communications]

Polygenic risk score for ulcerative colitis predicts immune checkpoint inhibitor-mediated colitisREVIEWERS' COMMENTS:

Reviewer #1 (Remarks to the Author): with expertise in cancer immunotherapy, genetics, clinical

General comments

This study investigates the association between genetic predisposition to UC /Crohn's disease and immune-mediated colitis in non-small cell lung cancer patients receiving immune checkpoint inhibitors (ICIs). The study develops and validates polygenic risk scores for UC and Crohn's disease in cancer-free individuals and assesses the role of these PRS in predicting colitis in a large cohort of ICI-treated lung cancer patients, finding that the PRSUC is modestly associated with risk of IMC. There is no association between PRSUC and PFS/OS. The authors also explore the association between human leukocyte antigen (HLA) alleles associated with UC risk and colitis. Overall, the authors are to be commended on this well-designed study utilising a large cohort to answer a question that is clinically relevant and addresses a knowledge gap in the field.

Minor comments

- Further detail in the methodology/results on generation of the polygenic risk scores could be provided to enhance reproducibility. Specify the selection criteria for variants included in the scores, number of variants included in each score and the degree of overlap between scores.
- Regarding the analyses of IMC on PFS/OS and PRSUC on PFS/OS – is it necessary to present two models here? Survival bias is a clear issue with any model including all the patients. Consider presenting only the 90-day landmark analysis throughout figures and tables to improve clarity
- The effect size of PRSUC on IMC is modest for both all-grade and severe IMC – the change in PRS is not a very strong predictor of IMC. Based on these results, it's hard to see this score having much clinical utility, especially given the low incidence of IMC in this population (4%). Comments in the discussion overstate the potential impact given the modest association between the score and IMC and authors should consider revising these comments.

- Can the authors defend the decision to correct for multiple hypothesis testing on HLA markers? There is strong biological plausibility, plus the analyses were already restricted to just 12 common HLA alleles known to be associated with UC. I recommend the authors should consider calling the association with HLA-DRB1*12:01 significant and review the ensuing comments in discussion. It seems perhaps it is being dismissed when actually the OR 2.63 suggests it might be a stronger predictor of IMC (although appreciate the confidence interval is wide)
- Table 3; layout could be clearer to better aid interpretation. Is it necessary to present both the cox model and the cox models with 90-day landmark? Consider just presenting the cox models with 90-day landmark, and then instead of listing different models, the row name can be the variable i.e. all-grade vs severe IMC, which would be a lot easier to read
- Recommend the association between PRS and PFS/OS is reported in a table/figure, so the reader can find it more easily e.g. Could be added to table 3 as a new section on the bottom.
- Introduction states that the incidence varies from 1-25% and then states that severe colitis leads to treatment cessation in 15-30% - please clarify these conflicting statements.
- Comment in abstract – “genetic susceptibility to CD/UC may predispose to IMC, but the link is poorly understood” – this implies there is already evidence linking these, needs to be referenced or explained further in the introduction
- Consider adding detail about the study type in the title e.g. indicating that it’s a PRS study and the approach used measuring risk of IBD
- Please correct typo in limitations; ancestry is misspelled

Reviewer #2 (Remarks to the Author): with expertise in statistical genetics

Authors conducted the association study of the polygenic risk scores (PRS) with immune-mediated colitis (IMC) after immune checkpoint inhibitors (ICI). PRS of ulcerative colitis (UC) and Crohn's disease (CD) were assessed with the IMC results in ~1,300 patients with non-small cell lung cancer (NSCLC). PRS_UC showed nominal associations but PRS_CD did not. The HLA gene did not show significant association. While IMC are known to be associated with survival, PRS did not show associations with the survivals. While this study handles a potentially interesting and clinically important topic, the results were nominal and lack robustness in conclusion.

1. Associations of PRS_UC is nominal. Given the relatively low frequency of the IMC events, further validation using an independent cohort is necessary.
2. PRS can have different values based on the calculation methods and the discovery GWAS results. Sensitivity tests for such parameters were not conducted.
3. PRS calculations for a wider range of human complex traits are necessary.
4. It is not clear whether the suggested findings were specific to NSCLC or not.

Reviewer #3 (Remarks to the Author): with expertise in genetics, immunogenomics

This manuscript describes an effort to establish and evaluate a genetic risk score for the prediction of developing immunotherapy-mediated colitis, a common adverse event during immune checkpoint blockade therapy. Since the advent of immunotherapy as a novel and promising therapeutic approach for late stage tumors, therapy-induced adverse events have been a serious problem. The possibility to predict the individual risk for developing such adverse events would be a great advantage that could inform therapy decisions and preventive care.

The goal of the present study is therefore of great importance and very timely. However, it is also widely known that genetic risk scores for complex diseases are inherently inaccurate, so that their true applicability in a clinical context remains very limited.

Developing genetic risk scores requires significant patient numbers, a general problem with immunotherapy as patient numbers are still relatively small and properly documented adverse events are even fewer. The authors address this problem with an elegant approach, developing the risk score based on disease with similar symptoms but which is much more common: inflammatory bowel disease (IBD).

Eventually, using significant IBD patient cohorts, the authors show that a risk score developed for ulcerative colitis also predicts the risk for immunotherapy-mediated colitis.

This is a very interesting finding, even if the accuracy of the scores remains limited.

Overall, this study sheds interesting light on potential similarities in genetic associations between UC and IMC, and provides a promising avenue towards using genetic risk scores in the immunotherapy context.

Below I'm detailing a few comments that I think should be addressed to improve the manuscript.

Major:

Results:

Although the 'GeRi cohort' is described in the methods section, please provide a few more details already in the results. For instance:

„The cohort comprised approximately 50% men and the mean age at diagnosis was 65 years“ Diagnosis of what? Probably cancer, but please state this precisely.

Also where this cohort was collected, whether it includes different ethnicities and whether this cohort has previously been published.

What do you mean with 'adjusted PRS-UC'? Is this the RSC-UC score from the 70% training data mentioned just before, or was this 'adjusted' further ?

This is not exactly the goal of the current study, but I think it would be very illuminating

towards the genetics and etiology of the two IBD types: Could you describe how the two PRSs correlate with each other, and how they predict the other disease type? What is the AUROC when using PRS-UC to predict CD, and vice versa?

There should be some shared genetic variants, but also some differences between CD and UC, and reporting these analyses might shed light on this relationship.

HLA imputation and analysis: The authors state in the methods that HLA alleles were imputed to 2-digit resolution, but they show HLA alleles at 4-digit (2-field) resolution in suppl figure 4. Please clarify.

Also, it should be stated whether HLA imputation was in any way validated. Was there at least a subset of individuals with known HLA genotype that could be used for evaluating the imputation accuracy? Otherwise it remains uncertain whether the lack of an association signal between HLA and IMC is true or just a result of poor HLA genotype inference. If there is no validation, this problem should also be mentioned and discussed.

Include a discussion of the weaknesses of using PRS in general and the ones proposed here specifically! How realistic is it that they would really be used in the clinical setting given their limited accuracy? Also address the point that PRSs might not work the same for all ethnicities, especially in the light of Degenhardt et al. 2021 (doi: 10.1093/hmg/ddab017) showing ethnicity-specific genetic associations for UC.

Minor:

Line/page numbering would have been helpful for reviewing!

Introduction: The statements that IMC occurs in 1%-25% of ICI patients and that hospitalization because of IMC occurs in ,up to 15-30%' of ICI patients do not seem to match (how can 15% be hospitalized because of IMC if only 1% get it). Please revise.

We appreciate the comments and suggestions from the Reviewers to improve our research article. We have addressed all comments point by point. Most importantly, based on the recommendations by reviewer 2, we have added data from another independent cohort of ICI-treated patients treated at VUMC to replicate our findings. Using the combined datasets, we can also make more accurate assessments about the ability of PRS_{UC} to predict immune checkpoint inhibitor-mediated colitis (IMC) in the patients with higher risk of IMC such as patients treated with combination ICI therapy.

All questions and comments by the Reviewers (black text) were answered (blue text) on a point-by-point basis with references made to the location of the changes in the manuscript (Page, Line) if relevant. These, as well as additional changes and corrections to the manuscript are tracked in red for easy identification (Manuscript_markup).

Reviewer #1 (Remarks to the Author): with expertise in cancer immunotherapy, genetics, clinical

General comments

This study investigates the association between genetic predisposition to UC /Crohn's disease and immune-mediated colitis in non-small cell lung cancer patients receiving immune checkpoint inhibitors (ICIs). The study develops and validates polygenic risk scores for UC and Crohn's disease in cancer-free individuals and assesses the role of these PRS in predicting colitis in a large cohort of ICI-treated lung cancer patients, finding that the PRS_{UC} is modestly associated with risk of IMC. There is no association between PRS_{UC} and PFS/OS. The authors also explore the association between human leukocyte antigen (HLA) alleles associated with UC risk and colitis. Overall, the authors are to be commended on this well-designed study utilising a large cohort to answer a question that is clinically relevant and addresses a knowledge gap in the field.

Minor comments

- Further detail in the methodology/results on generation of the polygenic risk scores could be provided to enhance reproducibility. Specify the selection criteria for variants included in the scores, number of variants included in each score and the degree of overlap between scores. We have included additional details about the selection criteria, number of variants in each score (Page 13, Lines 390-414)

There is some degree of overlap between the genetics of ulcerative colitis and Crohn's disease, and we have added details to reflect that in the Results section (Page 4-5, Lines 132-137).

- Regarding the analyses of IMC on PFS/OS and PRS_{UC} on PFS/OS – is it necessary to present two models here? Survival bias is a clear issue with any model including all the patients. Consider presenting only the 90-day landmark analysis throughout figures and tables to improve clarity.

We are now only presenting cox proportional hazards models with 90-day landmark for IMC and PFS/OS analysis and PRS and PFS/OS analysis. We have updated the text (Page7, Lines 220-224; Page 10, Lines 312-316, Page 16, Lines 487-496), figures (Figure 3 and Supplementary Figure 5) and table (Table 4)

- The effect size of PRS_{UC} on IMC is modest for both all-grade and severe IMC – the change in PRS is not a very strong predictor of IMC. Based on these results, it's hard to see this score having much clinical utility, especially given the low incidence of IMC in this population (4%).

Comments in the discussion overstate the potential impact given the modest association between the score and IMC and authors should consider revising these comments
We agree that the risk of IMC is relatively low and that the PRS is relatively modest overall. However, based on combined analyses with the replication data (see comments to reviewer 2), we can now offer more precise estimates of the effect for severe irAEs (Odds ratio [OR] per standard deviation [SD]: 1.49, 95% CI =1.18-1.88, $p = 9 \times 10^{-04}$). Furthermore, among the persons on dual therapy who have a much higher absolute risk (incidence 15-20%), we found an even stronger effect size for the severe irAEs (OR per SD = 2.20, 95% CI = 1.07 = 4.53, $p = 0.03$). Therefore, we believe our findings are clinically relevant and have an impact particularly in the treatment group (dual therapy).

- Can the authors defend the decision to correct for multiple hypothesis testing on HLA markers? There is strong biological plausibility, plus the analyses were already restricted to just 12 common HLA alleles known to be associated with UC. I recommend the authors should consider calling the association with HLA-DRB1*12:01 significant and review the ensuing comments in discussion. It seems perhaps it is being dismissed when actually the OR 2.63 suggests it might be a stronger predictor of IMC (although appreciate the confidence interval is wide).

While there is indeed strong biological plausibility for the association, and the analysis focused on a limited set of 12 HLA markers linked to UC, applying corrections for multiple testing helps mitigate the risk of false-positive results and enhances the reliability of the findings. We have reported both the nominal associations and the multiple hypothesis test corrected results which will allow others to interpret our results.

- Table 3; layout could be clearer to better aid interpretation. Is it necessary to present both the cox model and the cox models with 90-day landmark? Consider just presenting the cox models with 90-day landmark, and then instead of listing different models, the row name can be the variable i.e., all-grade vs severe IMC, which would be a lot easier to read
We agree with the reviewer and have revised Table 3.

- Recommend the association between PRS and PFS/OS is reported in a table/figure, so the reader can find it more easily e.g. Could be added to table 3 as a new section on the bottom.
We have added the results from PRS and PFS/OS analysis to Table 5.

- Introduction states that the incidence varies from 1-25% and then states that severe colitis leads to treatment cessation in 15-30% - please clarify these conflicting statements.
The incidence of immune checkpoint inhibitor-mediated colitis and severe immune checkpoint inhibitor-mediated colitis varies based on type of therapy. We have revised the text in our Introduction (Page 3, Lines 90-93).

- Comment in abstract – “genetic susceptibility to CD/UC may predispose to IMC, but the link is poorly understood” – this implies there is already evidence linking these, needs to be referenced or explained further in the introduction.

The clinical and endoscopic presentation of immune checkpoint inhibitor-mediated colitis in patients treated with immune checkpoint inhibitors resembles inflammatory bowel disease¹. Furthermore, histologically, some of the findings among immune checkpoint inhibitor-mediated colitis resemble inflammatory bowel disease². Therefore, our hypothesis was that there may be shared genetics between different forms of inflammatory bowel disease and immune checkpoint inhibitor-mediated colitis. We have updated the text both in the abstract (Page 2, Lines 49-51) and introduction to reflect this (Page 3, Lines 93-96).

1. Chen JH, Pezhouh MK, Lauwers GY, Masia R. Histopathologic Features of Colitis Due to Immunotherapy With Anti-PD-1 Antibodies. *The American Journal of Surgical Pathology*. 2017;41(5):643-654. doi:10.1097/PAS.0000000000000829
2. Nahar KJ, Rawson RV, Ahmed T, et al. Clinicopathological characteristics and management of colitis with anti-PD1 immunotherapy alone or in combination with ipilimumab. *J Immunother Cancer*. 2020;8(2):e001488. doi:10.1136/jitc-2020-001488

- Consider adding detail about the study type in the title e.g. indicating that it's a PRS study and the approach used measuring risk of IBD

We have revised the title to "Polygenic risk score for ulcerative colitis predicts immune checkpoint inhibitor-mediated colitis".

- Please correct typo in limitations; ancestry is misspelled

We have corrected the error (Page 10, Line 307).

Reviewer #2 (Remarks to the Author): with expertise in statistical genetics

Authors conducted the association study of the polygenic risk scores (PRS) with immune-mediated colitis (IMC) after immune checkpoint inhibitors (ICI). PRS of ulcerative colitis (UC) and Crohn's disease (CD) were assessed with the IMC results in ~1,300 patients with non-small cell lung cancer (NSCLC). PRS_UC showed nominal associations but PRS_CD did not. The HLA gene did not show significant association. While IMC are known to be associated with survival, PRS did not show associations with the survivals. While this study handles potentially interesting and clinically important topics, the results were nominal and lack robustness in conclusion.

1. Associations of PRC_UC is nominal. Given the relatively low frequency of the IMC events, further validation using independent cohort is necessary.

We have worked with another research group who have analyzed an independent cohort at VUMC of 873 patients on immune check point inhibitors. Using their data, we have replicated the results from our analysis (Page 6, Lines 170-193, Table 3). Furthermore, we also performed a meta-analysis between the two studies (Page 6-7, Lines 195-210, Table 3). The associated methods (Page 15-16, Lines 445-476) and results (Page 6-7, Lines 170-210, Table 3) from replication and meta-analysis are added to the manuscript. Overall, we believe that the new data both strengthen the confidence of our original conclusions and also allow us to make more precise estimates of the effects among patients receiving combination ICI therapy who have a very high risk overall and in whom the PRS_{UC} is strongly predictive (OR_{meta} per SD = 2.20, 95% CI = 1.07-4.53, $p = 0.03$).

2. PRS can have different values based on the calculation methods and the discovery GWAS results. Sensitivity test for such parameters were not conducted.

We calculated associations between IMC and several other PRSs for UC and all of them showed consistent association with HR per SD ranging from 1.24 to 1.41 (Page 5-6, Lines 166-168, Supplementary Table 2).

3. PRS calculations for a wider range of human complex traits are necessary.

This study was formulated with a specific hypothesis based on the observed clinical and histological similarities between autoimmune colitis such as Crohn's disease and ulcerative colitis and immune checkpoint inhibitor-mediated colitis seen in patients treated with immune checkpoint inhibitors^{1,2}. Therefore, we restricted this study to the traits that closely resemble the phenotype. We based this on our prior successful prediction of autoimmune thyroiditis³ and from

other investigators' prediction of skin irAEs⁴. Testing additional PRSs may be of interest but would also expand the multiple hypothesis testing correction required and, therefore, decrease power to find true associations.

1. Chen JH, Pezhouh MK, Lauwers GY, Masia R. Histopathologic Features of Colitis Due to Immunotherapy With Anti-PD-1 Antibodies. *The American Journal of Surgical Pathology*. 2017;41(5):643-654. doi:10.1097/PAS.0000000000000829
2. Nahar KJ, Rawson RV, Ahmed T, et al. Clinicopathological characteristics and management of colitis with anti-PD1 immunotherapy alone or in combination with ipilimumab. *J Immunother Cancer*. 2020;8(2):e001488. doi:10.1136/jitc-2020-001488
3. Luo J, Martucci VL, Quandt Z, et al. Immunotherapy-Mediated Thyroid Dysfunction: Genetic Risk and Impact on Outcomes with PD-1 Blockade in Non–Small Cell Lung Cancer. *Clinical Cancer Research*. 2021;27(18):5131-5140. doi:10.1158/1078-0432.CCR-21-0921
4. Khan Z, Di Nucci F, Kwan A, et al. Polygenic risk for skin autoimmunity impacts immune checkpoint blockade in bladder cancer. *Proceedings of the National Academy of Sciences*. 2020;117(22):12288-12294. doi:10.1073/pnas.1922867117

4. It is not clear whether the suggested findings were specific to NSCLC or not. The initial analyses were conducted in non-small cell lung cancer patients on immune checkpoint inhibitor therapy. However, our replication was conducted in a pan-cancer cohort of patients on immune checkpoint inhibitor therapy. Our replication and meta-analyzed results suggest that the immune checkpoint inhibitor-mediated colitis is most likely a result of the type of immunotherapy rather than type of cancer, which we have shown in our therapy-stratified analysis (Page 10, Lines 298-300, Table 2:discovery and Table 3: replication and meta-analysis)

Reviewer #3 (Remarks to the Author): with expertise in genetics, immunogenomics

This manuscript describes an effort to establish and evaluate a genetic risk score for the prediction of developing immunotherapy-mediated colitis, a common adverse event during immune checkpoint blockage therapy. Since the advent of immunotherapy as a novel and promising therapeutic approach for late-stage tumors, therapy-induced adverse events have been a serious problem. The possibility to predict the individual risk for developing such adverse events would be a great advantage that could inform therapy decisions and preventive care. The goal of the present study is therefore of great importance and very timely. However, it is also widely known that genetic risk scores for complex diseases are inherently inaccurate, so that their true applicability in a clinical context remains very limited.

Developing genetic risk scores requires significant patient numbers, a general problem with immunotherapy as patient numbers are still relatively small and properly documented adverse events are even fewer. The authors address this problem with an elegant approach, developing the risk score based on disease with similar symptoms but which is much more common: inflammatory bowel disease (IBD). Eventually, using significant IBD patient cohorts, the authors show that a risk score developed for ulcerative colitis also predicts the risk for immunotherapy-mediated colitis. This is a very interesting finding, even if the accuracy of the scores remains limited.

Overall, this study sheds interesting light on potential similarities in genetic associations between UC and IMC, and provides a promising avenue towards using genetic risk scores in the immunotherapy context.

Below I'm detailing a few comments that I think should be addressed to improve the manuscript.

Major:
Results:

Although the 'GeRI cohort' is described in the methods section, please provide a few more details already in the results. For instance: "The cohort comprised approximately 50% men and the mean age at diagnosis was 65 years" Diagnosis of what? Probably cancer, but please state this precisely.

Also, where this cohort was collected, whether it includes different ethnicities and whether this cohort has previously been published.

We have added additional details on the cohort (Page 4, Lines 116-118, Table 1). Our study comprised 1,316 non-small cell lung cancer patients treated with immune checkpoint inhibitor therapy. Detailed information about the participating sites is included in the Methods (Page 11, Lines 325-351, Table 1). In the results, we have mentioned that the cohort included four sites (Page 4, Line 115 and Page 11, Lines 326-351). We have included the type of cancer on Page 4, Line 116. Additionally, we have added numbers on self-reported race (overall and by race) in Table 1. We have also added information about the second cohort of VUMC patients we added in replication (Page 6, Lines 170-178 and Page 15, Lines 445-461, Supplementary Table 1).

What do you mean with 'adjusted PRS-UC'? Is this the RSC-UC score from the 70% training data mentioned just before or was this 'adjusted' further ?

Adjusted PRS_{UC} refers to the association between PRS_{UC} and UC adjusted for age at diagnosis for cases and age at enrollment for controls, sex, genotyping array, and the top 10 genetic ancestry principal components (PCs). We have updated the sentence on Page 4, Line 129).

This is not exactly the goal of the current study, but I think it would be very illuminating towards the genetics and etiology of the two IBD types: Could you describe how the two PRSs correlate with each other, and how they predict the other disease type? What is the AUROC when using PRS-UC to predict CD, and vice versa?

There should be some shared genetic variants, but also some differences between CD and UC, and reporting these analyses might shed light on this relationship.

We observed a modest correlation between the two PRSs (Pearson correlation = 0.38). Additionally, the AUROC for PRS_{UC} on CD was 0.58 (95% CI: 0.57 - 0.60), while PRS_{CD} on UC showed an AUROC of 0.58 (95% CI: 0.56 - 0.59). Based on these results, we can observe some shared genetic susceptibility between UC and CD. Nonetheless, the unique genetic factors underpinning each phenotype seem to be the predominant determinants of the individual PRS effects. We have added this to the manuscript (Page 4-5, Lines 132-137 and Page 8, Lines 250-255).

HLA imputation and analysis: The authors state in the methods that HLA alleles were imputed to 2-digit resolution, but they show HLA alleles at 4-digit (2-field) resolution in suppl figure 4.

Please clarify.

We have updated the terminology in the main text (Page 16, Line 480). We imputed HLA alleles to 2-field (4-digit) resolution.

Also, it should be stated whether HLA imputation was in any way validated. Was there at least a subset of individuals with known HLA genotype that could be used for evaluating the imputation accuracy? Otherwise, it remains uncertain whether the lack of an association signal between HLA and IMC is true or just a result of poor HLA genotype inference. If there is no validation, this problem should also be mentioned and discussed.

HLA imputation was not validated due to non-availability of HLA typing information. We have acknowledged this in the discussion of the manuscript Page 8 and Lines 244-247).

Include a discussion of the weaknesses of using PRS in general and the ones proposed here specifically! How realistic is it that they would really be used in the clinical setting given their limited accuracy? Also address the point that PRSs might not work the same for all ethnicities, especially in the light of Degenhardt et al. 2021 (doi: 10.1093/hmg/ddab017) showing ethnicity-specific genetic associations for UC.

We have elaborated on the potential weakness of using PRS and the generalizability issues that arise with the development of PRS in the discussion (Page 10 and Lines 302-310).

Minor:

Line/page numbering would have been helpful for reviewing!

We have added page and line numbers.

Introduction: The statements that IMC occurs in 1%-25% of ICI patients and that hospitalization because of IMC occurs in ,up to 15-30%' of ICI patients do not seem to match (how can 15% be hospitalized because of IMC if only 1% get it). Please revise.

The incidence of immune checkpoint inhibitor-mediated colitis and severe immune checkpoint inhibitor-mediated colitis varies based on type of therapy. We have revised the text in our Introduction (Page 3, Lines 90-93).

REVIEWERS' COMMENTS

Reviewer #1 (Remarks to the Author):

Comments have been satisfactorily addressed and the authors are to be congratulated on this improved version.

Reviewer #2 (Remarks to the Author):

Authors well responded the reviewer's comments. Validation in another cohort may strengthen the authors' findings.

Reviewer #3 (Remarks to the Author):

I appreciate the effort the authors have made in revising their manuscript. I think the manuscript has improved further. The author have addressed all my concerns except for one, which is the general (and specific) limitation of PRSs here. Genetic risk scores can always only predict the risk associated with genetic factors, but most complex disease have also major environmental risk effects. In my opinion, the authors of this (and any) study promoting genetic risk scores should address this point in their discussion in order to avoid false hope. Furthermore, in this particular study, the authors should also express numerically how accurate their risk prediction is. Even though it is statistically significant, its accuracy is very limited, and this should be made clear to any reader.

I leave it at the discretion of the editor to decide whether the authors should revise their manuscript in this light or not.

We appreciate the comments and suggestions from the Reviewers to improve our research article. We have addressed all comments point by point.

All questions and comments by the Reviewers (black text) were answered (blue text) on a point-by-point basis with references made to the location of the changes in the manuscript (Page, Line) if relevant. These, as well as additional changes and corrections to the manuscript, are tracked in red for easy identification.

REVIEWERS' COMMENTS

Reviewer #1 (Remarks to the Author):

Comments have been satisfactorily addressed, and the authors are to be congratulated on this improved version.

Reviewer #2 (Remarks to the Author):

Authors well responded the reviewer's comments. Validation in another cohort may strengthen the authors' findings.

We acknowledge that another validation in a separate cohort will strengthen these findings and should be considered in future studies. We have added text in the manuscript to reflect this (Page 9, Lines 300-303).

“Although we included one replication cohort, additional studies of more ICI-treated patients will help strengthen our findings and, in particular, may give better power to evaluate HLA associations and other individual loci that may improve our understanding of the genetic similarities and potential differences between IMC and UC.”

Reviewer #3 (Remarks to the Author):

I appreciate the effort the authors have made in revising their manuscript. I think the manuscript has improved further. The author have addressed all my concerns except for one, which is the general (and specific) limitation of PRSs here. Genetic risk scores can always only predict the risk associated with genetic factors, but most complex disease have also major environmental risk effects. In my opinion, the authors of this (and any) study promoting genetic risk scores should address this point in their discussion in order to avoid false hope.

Furthermore, in this particular study, the authors should also express numerically how accurate their risk prediction is. Even though it is statistically significant, its accuracy is very limited, and this should be made clear to any reader.

I leave it at the discretion of the editor to decide whether the authors should revise their manuscript in this light or not.

We agree with the point made by the reviewer and have added the importance of considering the environmental effects along with genetic effects in the discussion (Page 9, Line 303-309).

“For most complex traits including autoimmune disorders and, likely for irAEs, environmental factors also play an important role. Our study does not address how environmental factors affect the risk of IMC. For example, gut microbiome may modify susceptibility to and severity of IBD (PMID 33253681) and, therefore, may also contribute to the susceptibility of developing IMC in cancer patients who have undergone ICI treatment. To determine the joint associations between PRS_{UC} and environmental risk factors, further studies are necessary.”